# Application of mmWave Radar Sensor for People Identification and Classification

**DOI:** 10.3390/s23083873

**Published:** 2023-04-10

**Authors:** Xu Huang, Nitish Patel, Kit P. Tsoi

**Affiliations:** 1Department of Electrical and Computer Engineering, The University of Auckland, Auckland 1010, New Zealand; 2Faculty of Information Engineering, University of Shandong Ying Cai, Jinan 250104, China

**Keywords:** millimeter wave, radar, detection, identification, classification

## Abstract

Device-free indoor identification of people with high accuracy is the key to providing personalized services. Visual methods are the solution but they require a clear view and good lighting conditions. Additionally, the intrusive nature leads to privacy concerns. A robust identification and classification system using the mmWave radar and an improved density-based clustering algorithm along with LSTM are proposed in this paper. The system leverages mmWave radar technology to overcome challenges posed by varying environmental conditions on object detection and recognition. The point cloud data are processed using a refined density-based clustering algorithm to extract ground truth in a 3D space accurately. A bi-directional LSTM network is employed for individual user identification and intruder detection. The system achieved an overall identification accuracy of 93.9% and an intruder detection rate of 82.87% for groups of 10 individuals, demonstrating its effectiveness.

## 1. Introduction

Object identification, also called “who is where”, is a type of machine learning that trains algorithms to identify objects based on labeled data examples. This technique is crucial for developing smart services and applications in indoor spaces, such as occupancy detection, energy management, safety monitoring, and background music selection [1]. High precision and autonomy are essential for the seamless integration of these services.

Personal devices, such as ID cards, RFID badges, smartphones, or smart bracelets, have been used as identification methods. However, the requirement that the device is inseparable from the user limits their application in some scenarios.

To overcome the limitations of identification methods implemented using personal devices, device-free methods have become increasingly popular for object recognition in various scenarios. Image-based techniques, including cameras, radar, and Ranging Doppler Azimuth, are effective when the image provides a clear field of view [2], but using cameras raises privacy concerns in home and business environments [3]. To address these concerns, radio signal frequency methods, such as ambient WiFi signal variations, have been proposed to identify people based on their gait patterns [4,5]. However, these methods must demonstrate robustness in different environmental conditions, particularly indoor object identification. In addition, the radio signal frequency approach requires separate transmitter and receiver pairs and cannot distinguish between two walking individuals simultaneously. Point cloud data-based onboard sensors, such as LiDAR and depth cameras, have been proposed to identify and track multiple walking individuals [6]. Still, this approach is not cost-effective for general use and has a limited field of view and accuracy due to the depth of the LiDAR sensors.

This paper presents a novel system for the people identification method using millimeter-wave (mmWave) radar sensor data. An indoor people identification pipeline is proposed, combining radar point cloud data and a deep learning network through a commercial mmWave radar sensor. With limited works for the object classification method presented in [7,8,9], this paper improves identification accuracy by implementing a Long-Short Term Memory (LSTM) network and a random forest classifier algorithm.

The paper is organized as follows. Section 2 details object identification and classification research, including the various sensor technologies and algorithms. Section 3 presents the proposed architecture, system implementation details, and the experiment setup. Section 5 introduces the system configuration, followed by the evaluation and discussion of the results in Section 6. Finally, the study is summarized, and the potential future work is presented in Section 7.

## 2. Related Works

The use of various radar signal representations for object identification and classification has been reviewed in recent deep-learning algorithms in academic and commercial systems. One of the major challenges is selecting an appropriate representation for radar signals for input into different deep learning algorithms [2]. These representations include radar grid maps, Range-Doppler-Azimuth maps, micro-Doppler signatures, and radar point cloud data.

In one paper [10], a method for static object classification using radar grid maps was presented. The classification was performed using a CNN algorithm, with semantic knowledge trained using the generated radar occupancy grid. However, using the radar grip generation, this approach can result in the loss of static object intrinsic characteristics and added system workload due to the massive mapping with many empty pixels, potentially leading to unstable classification.

In another paper [11], a deep learning-based vehicle classification system was proposed using radar signals. The radar Range-Azimuth was used as a 3D tensor, followed by velocity data to create a complete Range-Velocity-Azimuth 3D radar tensor, which was then input into a Long Short-Term Memory network (LSTM). In another paper [2,12,13], the use of micro-Doppler features for object classification using different deep learning methods was studied. The high-resolution radars result in better-defined feature representations. However, these systems can only predict the presence or absence of an object from the radar Doppler spectrograms. The range and angle dimension information are inaccessible.

In another paper [14], point clouds obtained from high-resolution radar sensors were shown to represent 3D data that preserves geometric information in a 3D space. In another paper [7], a PointNet was improved to become a PointNet++ model [15]. The original PointNet was designed for processing 3D LiDAR point clouds. The PointNet++ model was adapted from the original 3D model to facilitate 2D object identification and bounding box estimations using radar point clouds. In another paper [16], a radar-only 3D object identification system based on deep learning was introduced and trained on a public radar dataset. This method improves the training process by introducing additional labels on the radar datasets. This reduces overfitting problems in deep learning training. In another paper [17], a novel data transformation structure from LiDAR point clouds to the radar-like points dataset was presented and then applied to Complex-YOLO for 3D object detection.

## 3. System Design

A system for object identification using point cloud data generated by a millimeter-wave (mmWave) radar sensor is presented in this section. The object’s location and identity are inferred through a deep recurrent neural network trained with a user-known database. The pipeline for the system is shown in Figure 1.

The point cloud data are generated from range FFT, Doppler-FFT, and angle estimation and packaged into a TLV data format frame. The next step is to detect different objects by merging points into clusters using a modified version of the DBSCAN clustering algorithm. The identified objects are then recognized by deploying a deep recurrent neural network that analyzes the time-sequential data from the mmWave radar sensor. Finally, a Random Forest classifier is trained to determine whether the input category is an insider or intruder based on the feature distance to centers, calculated with the co-supervision of softmax-loss and center-loss.

### 3.1. Point Cloud Generation and Static Clutter Removal

The point cloud data are packaged into a data frame using the TLV format. Hence, a parsing section must ensure reliable and accurate extraction before the data analysis. Each round of data parsing begins by reading the frame header into an array containing information such as the packet length, frame number, number of TLV’s (number of data points in the point cloud), etc. Then the TLV data, which contains the point cloud data, is read into another array. The TLV’s size depends on the points detected in the field of view. The TLV header contains the TLV length, which is used to read the values by indexing the correct positions of the data frame.

Static clutter removal was designed to eliminate as many static points as possible, which are non-range changing (static) objects from the scene. The steps of the static clutter removal algorithm are listed as follows.

Step 1: Range processing performs Fast Fourier Transform (FFT) on Analog to Digital Converter (ADC) samples per antenna per chirp. FFT output is a set of range bins;

Step 2: Perform static clutter removal by subtracting the estimated Direct Current (DC) component from each range bin;

Step 3: Range processing results in local scratch buffers are Enhanced Direct Memory Access (EDMA) to the radar data cube with transpose.

### 3.2. Clustering

The detected targets are merged into clusters using the DBSCAN algorithm [18] to determine which points in the scene at a given time are reflected by which targets. However, the density of collected data can vary with time and distance from the sensor, leading to a decrease in accuracy due to the variable density issue [19]. To address this issue, an improved density-based clustering algorithm [3] developed in our previous work that can handle variable cluster densities is used to merge points into targets.

Due to inconsistencies in the scattering points’ vertical (Z) axis, projecting a 3D coordinates dataset into a 2D (X-Y) plane coordinates are often challenging, as shown in Figure 2. Therefore, an improved density-based clustering algorithm [3] is imposed on the dataset, such that the Euclidean distance equation is improved by putting less weight on the Z-axis in the clustering algorithm as Equation (Equation 1) below.
(1)$$D^{2}\left(p^{i},p^{j}\right)={\left(p_{x}^{i}-p_{x}^{j}\right)}^{2}+{\left(p_{y}^{i}-p_{y}^{j}\right)}^{2}+\alpha {\left(p_{z}^{i}-p_{z}^{j}\right)}^{2}$$
where $p^{i}$ and $p^{j}$ are two points at different location and α regulates the weight of Z-axis distance.

### 3.3. Tracking

The location and trajectory of targets are obtained by processing continuous input point cloud data in the tracking stage. Data association of detection and a filter are necessary to increase the system’s robustness and handle signal attenuation. The centroids detected are linked to new tracks for multiple objects from frame to frame, either from incoming frames or unassociated with existing tracks. The data association is carried out using the simplified Global Nearest Neighbour (GNN) Algorithm, and each track undergoes a life cycle of continuous frames. During maintenance, the tracks may be marked inactive and deleted from the existing list. A recursive Kalman Filter is employed to predict and correct tracks, but this paper does not cover the data association; the details of the Kalman Filter can be found in [19].

### 3.4. Identification

The correct points corresponding to the tracked objects are identified with the detection and tracking steps, and their identities can be recognized. Given that the objects are enclosed within a boundary box with the points inside, the objects are voxelized into an occupancy grid in each trajectory frame. The occupancy grid contains human body shape information, such as the center of mass, providing a distinctive feature for different human heights. Combined with motion features, such as gait and body shape information, the identity of a tracked object can be determined by feeding the sequential occupancy grids into a classifier. However, direct intrinsic feature extraction from the occupancy grids can be challenging, which may lead to ineffective point cloud data classification [20].

This paper proposed a long short-term memory (LSTM) approach as an identity classifier. Given that the LSTM network is a recurrent neural network (RNN) predominantly used to learn, process, and classify sequential data between time steps, it automatically allows the network to extract and identify the features through its training process. First, each frame of the point cloud data is flattened and converted into a feature vector. The feature vectors are later passed into a bi-directional LSTM network followed by a dense layer, forming a sequence processing model. The processing model contains two LSTMs. One LSTM takes the input forward, and the other LSTM takes the input backward. The BI-LSTM results are fed into its dense layer, and the corresponding result is later passed into a softmax layer to retrieve the classification results, as shown in Figure 3.

*T* represents the number of data frames. The lowercase *k* represents the number of people, and the capital *K* in the bracket represents layer sizes.

### 3.5. Intruder Detection

Intruder detection classifies out-of-set samples as intruders. The softmax layer in the output layer of a neural network predicts the intruders by estimating a multinomial probability distribution. However, scattered features learned from the softmax loss can negatively impact the ability to differentiate samples from external intruders. To overcome this, the center loss is combined with the softmax loss to form deep features [21,22]. Sample features are extracted from the deep neural network and then used to minimize the intra-class distance through the center loss, resulting in clustering features belonging to the same class.

The distance between an intruder’s extracted features and the inner cluster’s feature center should be greater than a threshold in the training set. The combination of softmax and center loss is defined as Equation (Equation 2).
(2)$$\ell =\ell_{softmax}+\lambda \ell_{center}$$
where λ is the weight for both the softmax and center losses, it balances the two loss functions and dominates the intra-class variations. We varied λ from 0 to 0.1 to learn different models. The corresponding softmax and center losses are defined as Equation (Equation 3).
(3)$$\begin{array}{cc} \; & \ell_{softmax}=-\sum_{i=1}^{m}\log\frac{e^{W_{y_{i}}^{T}f_{i}}+b_{y_{i}}}{\sum_{j=1}^{g}e^{{W_{j}}^{T}}f_{i}+b_{j}},\\ \; & \ell_{center}=\sum_{i=1}^{m}∥f_{i}-c_{y_{i}}{∥}_{2}^{2} \end{array}$$
where *g* represents the training data of *m* samples from *g* objects ($y_{i}$∈1,...,g). $f_{i}$∈$R^{n}$ represents the *i*th extracted feature, and $c_{y_{i}}$∈$R^{n}$ represents the center of *i*th feature label $y_{i}$. *W* and *b* are learnable weights and biases of the current network.

A trained classification network is shown in Figure 4. A dense layer is added after the bi-directional LSTM network to resolve the issue of features not converging with center loss. An embedding layer is then used to label the object’s ID to the center of the features. After the squared Euclidean distance calculation, the result is the center loss.

The distances between the extracted features of an intruder and the centers of the features of the inner cluster are calculated. The sample is classified as an intruder when these distances exceed a set threshold in the training data. The Random Forest layer then makes a binary prediction to identify whether the sample is an intruder or an insider, based on these varying distances to the feature centers.

## 4. System Implementation

### 4.1. Experiment Setup

The proposed object identification system comprises a radar sensor and a laptop control terminal. The radar sensor used for the evaluation was the IWR6843ISK, a 60 GHz commercial mmWave sensor kit [23]. The point cloud data obtained by the radar was collected in a laboratory environment and cross-verified with an HD camera tracking system, which provided the ground truth identities and marked positions of users and intruders. The 3D point cloud data were transferred from the radar to the laptop terminal for further processing.

The ground truth training and testing samples were manually labeled for their identities. The deep learning network classifier was developed using the Keras and Tensorflow libraries. The system’s performance was evaluated by comparing the predicted results with the ground truth data. The experiment setup is illustrated in Figure 5.

### 4.2. Data Collection

Data were collected from 12 participants in a general electronics laboratory, where each participant walked randomly in a designated area for 20 min. The participants’ heights ranged from 167 cm to 186 cm, and varied body shapes were considered to model real indoor conditions accurately.

## 5. System Configuration

### 5.1. Sensor Setup

The IWR6843ISK sensor, a 60 GHz mmWave device, was used in the system with a frequency range of 60 GHz to 64 GHz and 4 GHz available bandwidth. The sensor had three transmitter antennas and four receiver antennas that generated point cloud data with a chirp cycle configuration of 173.6 μs and a slope of 54.725 MHz/μs. This resulted in a resolution of 0.084 m and a maximum range of 7.2 m. The sensor’s maximum radial velocity measurement was 8.38 m/s with a resolution of 0.17 m/s, and it was set to transmit 288 chirps per frame with a sampling rate of 20 frames per second.

### 5.2. Clustering Parameter Configuration

The configuration of the minClusterSize and maxDistance parameters for the clustering algorithm was optimized by heuristic testing of various combinations of the parameters using the dataset. A minClusterSize of 10 points and a maxDistance of 0.8 m were selected.

### 5.3. Classifier Training Configuration

The input data frame was flattened to a dimension of 16,000 in the first layer. The Adam optimizer was used with 128 hidden Bidirectional LSTM (BiLSTM) units and a dropout ratio of 0.5. The dataset was divided into 90% training data and 10% testing data. The data were moved and rotated along the X and Y axes to prevent overfitting. The model was trained for 32 epochs.

### 5.4. Intruder Detection Configuration

The configuration of the dense feature layer in the classification network was set to 64 units for feature extraction, as depicted in Figure 4. To prevent overfitting and maintain performance, the weight of the softmax and center losses was chosen to be 1:0.5. The dropout rate was set to 0.5 for the Bidirectional LSTM and feature extraction layers to improve performance and reduce overfitting. The model was then trained for 32 epochs. The configuration for the Random Forest Classifier was determined through a combination of *k*-fold cross-validation (*k* = 5) and grid search. Where *k* is the number of folds within the dataset, and the base Random Forest training model is used as follows:



## 6. System Evaluation

### 6.1. Tracking

The accuracy of the proposed tracking system was compared with the Kinect v2 camera [24]. Both systems were set up in the laboratory environment, and the ground truth was obtained through HD camera tracking of marked user and intruder positions. Results of the comparison between the tracking trajectories and RMSE of the two systems are displayed in Figure 6 and Table 1. The proposed system had an extensive tracking range of 6 m compared to the Kinect v2’s 4.2 m. The proposed system had an RMSE of 0.2665 m, while the Kinect v2 had an RMSE of 1.025 m.

### 6.2. Identification

The challenge of analyzing human gait using radar-generated point cloud data with traditional vision-based methods is that the scattered nature of the cloud data makes it challenging to identify body parts [25,26]. To address this challenge, the Bi-directional LSTM was selected for feature extraction during model training because it has been shown to effectively model the rich temporal correlations in a long sequence of frames from both forward and backward directions and to outperform other architectures in terms of convergence speed and accuracy [27,28].

The BiLSTM, LSTM, and CNN architectures were compared to determine the most suitable neural network for the identification application. The models were trained for 32 epochs with a dropout ratio of 0.5. The BiLSTM and LSTM had 128 hidden units, while the CNN had two convolution layers with a max pooling layer.

The training accuracy and loss of the BiLSTM architectures are shown in Figure 7, and the accuracy differences between architectures are shown in Table 2.

To determine if overfitting occurs when training a machine learning model, cross-validation is used. Data are split into two parts: the training and validation sets. The training set is used to train the model, and the validation set is used to evaluate its performance. The model’s progress during training is monitored through metrics on the training set. In contrast, the ability of the model to make new predictions on unseen data is measured through metrics on the validation set.

The performance of the BiLSTM architecture was compared to that of the other two architectures, LSTM and CNN, during the training process. BiLSTM outperformed the other two regarding training accuracy and validation increase, indicating a better fit to the training set and a greater ability to predict new data. Thus, BiLSTM was determined to be the best-fit architecture for modeling correlations in long sequences of time-frame data from both forward and backward directions. The confusion matrix of the method is shown in Figure 8 to evaluate the identification performance. However, standard LSTM, as a forward direction network, was found to decrease identification performance due to its inability to process information from the beginning of a long sequence.

The confusion matrix displays the predicted class as rows and the actual class as columns. The diagonal cells indicate correctly classified observations, while the off-diagonal cells indicate incorrectly classified observations. Both the number and percentage of total observations are shown in each cell.

In the rightmost column of the plot, the percentage of all samples predicted to belong to each correctly and incorrectly classified class is displayed. These metrics are referred to as precision and false discovery rate. The bottom row shows the percentage of all samples belonging to each class that was correctly and incorrectly classified. The bottom right cell of the plot displays the system’s overall accuracy, with 93.9% accuracy and a ten-person identification accuracy of 93.6%.

### 6.3. Intruder Detection

The intruder detection was evaluated using the softmax loss for intruders prediction and center loss for features deep extraction. This enlarged the feature distances within the inner cluster, enabling the use of different numbers of people in the user dataset. Both user and intruder datasets were trained using a binary random forest classifier, with a 1:1 ratio of training and testing datasets.

Table 3 shows precision, recall, F1 score, and accuracy for various numbers of intruders. Precision is the ratio of actual intruders classified as intruders, recall is the percentage of intruders identified, and accuracy is the proportion of samples correctly categorized.

With an increase in the number of intruders, the representation of the detection model decreased. The increased number of intruders caused more samples to become occluded and scattered across the high-dimensional feature area, leading to misclassification. Future work aims to improve results by training larger datasets, enhancing the model’s ability to handle a larger number of intruders.

### 6.4. Discussion on Robustness

For the experiment setup, we set the sensor’s maximum range to 6 m. In principle, the range can be extended to 30 m. However, it will reduce spatial precision and cause a worse signal-to-noise ratio. Moreover, if the subject is too far from the sensor, detecting and identifying them from the sensor noise is challenging.

During the experiments, we found that the reflection of the mmWave sensor was affected by windows and mirrors. As a result, noisy objects occasionally appear when these surfaces are surrounded. However, the experiment results were not strongly influenced since the sites mainly had walls and one window. However, it is worthwhile to consider the impact of disturbing surfaces in a real-world deployment.

## 7. Conclusions

This paper proposes a highly accurate indoor people identification and classification system using mmWave radar technology. The scattered point cloud data were obtained from a commercial mmWave radar sensor, and human objects, and their trajectories were associated and tracked using clustered and extracted point clouds. The identities of the objects were then recognized by an optimized recurrent neural network, which was trained using a softmax layer and a center loss added to the softmax loss to address the scattered features problem. Finally, a binary output was predicted by a Random Forest layer to identify whether the sample was an intruder or an insider.

Experimental results indicated that the system’s overall identification accuracy reached 93.9%, with a ten-person intruder detection accuracy of 82.87%. To improve the model’s performance, and future work will involve using multiple mmWave sensors to increase the amount of points data in each frame and training larger datasets to handle more intruder detections.

## Figures and Tables

**Figure 1 sensors-23-03873-f001:**
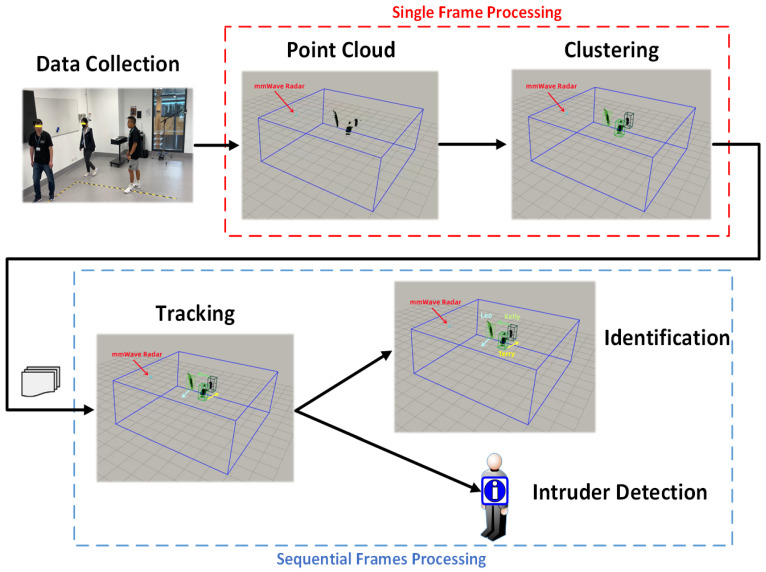
System Overview.

**Figure 2 sensors-23-03873-f002:**
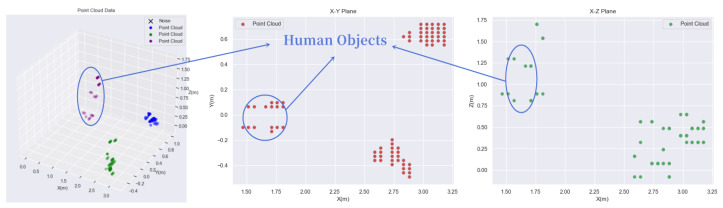
2D and 3D projections of the point cloud. The point corresponding to three human objects are compact in the horizontal X–Y plane but dispersed in the vertical X–Z plane.

**Figure 3 sensors-23-03873-f003:**
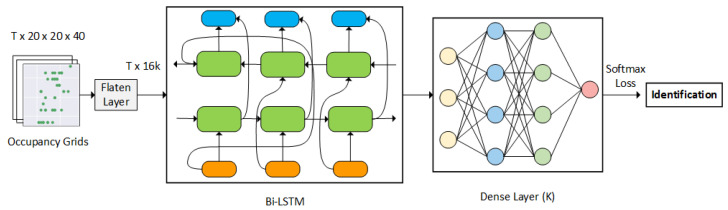
User classification structure.

**Figure 4 sensors-23-03873-f004:**
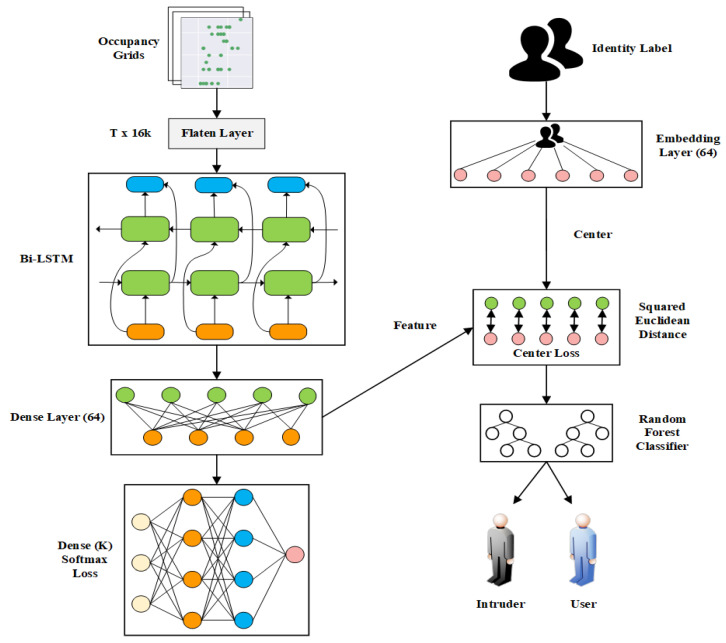
Classification Network Structure.

**Figure 5 sensors-23-03873-f005:**
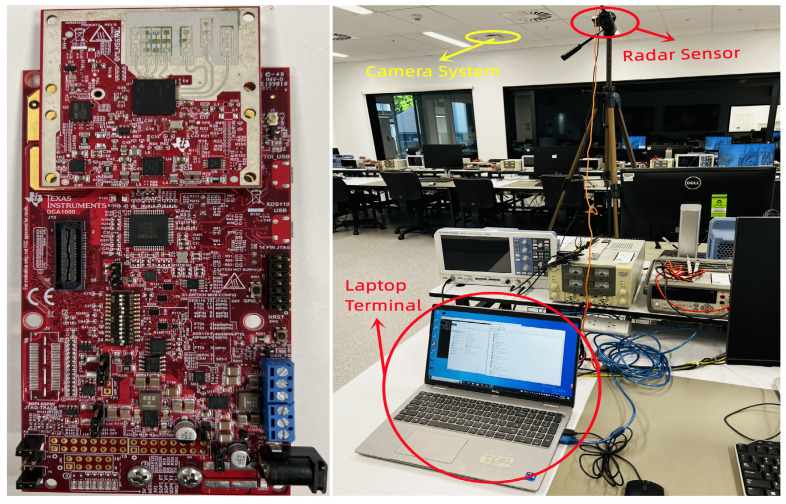
Experiment Setup. IWR6843 Sensor and Experiment Setting.

**Figure 6 sensors-23-03873-f006:**
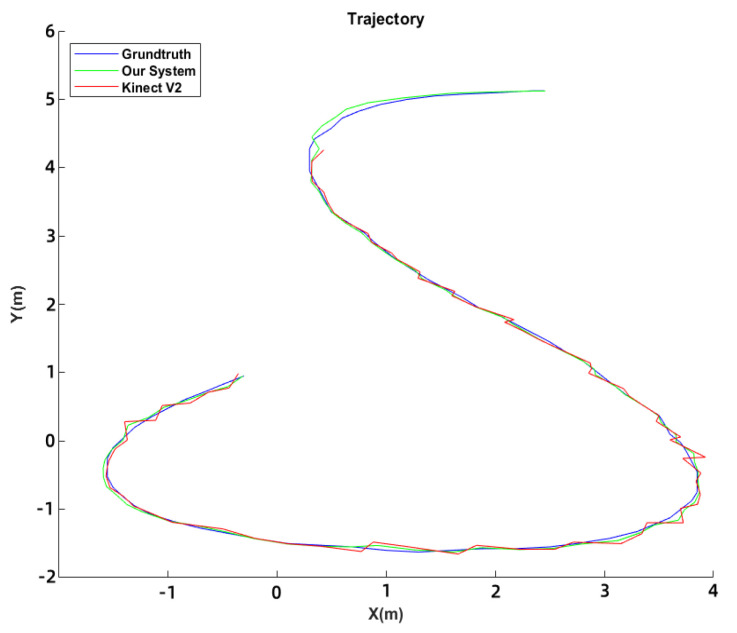
Tacking Trajectory.

**Figure 7 sensors-23-03873-f007:**
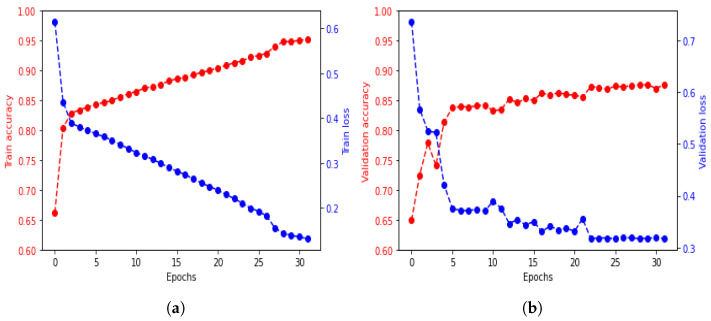
Train Accuracy/Loss of the BiLSTM. (**a**) BiLSTM Training Accuracy. (**b**) BiLSTM Validation Accuracy.

**Figure 8 sensors-23-03873-f008:**
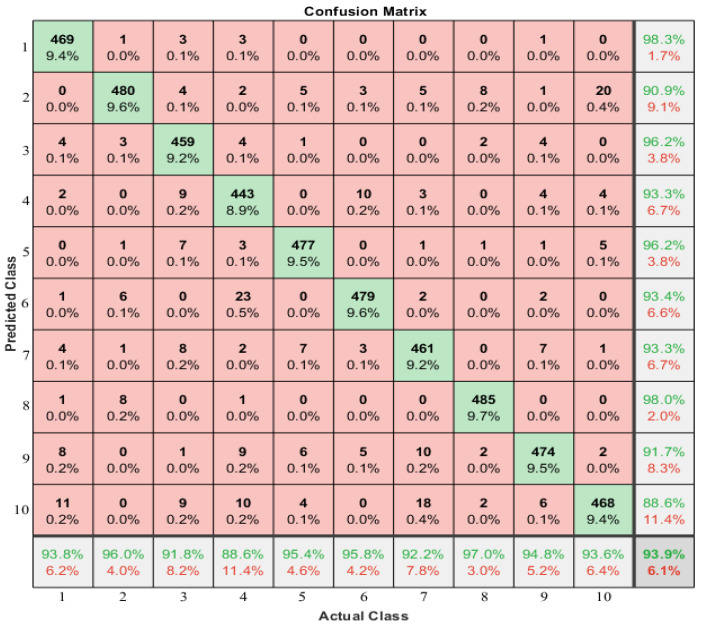
Confusion Matrix of 10 Users.

**Table 1 sensors-23-03873-t001:** Tracking Error Comparison Between Kinect v2 and Proposed System.

System	Total Frames	Object Number	RMSE
Kinect v2	5611	1	1.025
Proposed System	5611	1	0.2665

**Table 2 sensors-23-03873-t002:** The Accuracy Differences Between Architectures.

Architectures	Frames	Accuracy	Loss	Validation Accuracy	Validation Loss
BiLSTM	12,490	0.9522	0.1308	0.8763	0.3218
LSTM	12,490	0.9241	0.1704	0.8581	0.3497
CNN	12,490	0.8819	0.2068	0.8391	0.3518

**Table 3 sensors-23-03873-t003:** The Detection accuracy for different numbers of intruders.

Intruders	Precision	Recall	F1	Accuracy
2	0.9352	0.9863	0.9522	0.9681
4	0.8719	0.9581	0.9080	0.9278
6	0.8290	0.9021	0.8570	0.8825
8	0.8068	0.8941	0.8430	0.8720
10	0.7879	0.8777	0.8319	0.8287

## Data Availability

Not applicable.

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
