# Peer review of "Application of mmWave Radar Sensor for People Identification and Classification"

_sensors, 2023, doi:10.3390/s23083873_

Round 1
Reviewer 1 Report
Corrections and concerns in the text
44 2. Relate Related Works
45 We reviewed review recent deep-learning algorithms
,102 alpha (looks like “a” and not alpha) are two points at different location and a regulates the weight of Z-axis distance.
109 track. Frame-by-frame data association is based on the simplified GNN Algorithm (define GNN).
126 is flattened and converted into a feature vector. Then, it passed into a bi-directional LSTM network
127 followed by a dense layer, a sequence processing model consisting of two LSTM (why bi-directional LSTM?)
128 forward and the other backward (why forward and backward is needed?).
Figure 3: T represents the number of data frames, and K represents the number of people and layer sizes (why should K be the same for number of people and layer sizes?).
141 where λ is the weight for the two losses (how is lambda determined?).
180 frame is inputted input to the first layer to flatten the vector to a dimension of 16000.
191 This model was trained at 0.5 dropout rates and 32 epochs. The Random Forest Classifier configuration (what are the RF dimensions. That is, number of trees, depth etc?)
225 and backward. However, the standard LSTM is a feedforward network (No – the LSTM is NOT a feedforward network)
221 In contrast to table Table 2.
246 shown in table Table 3.
Major Concerns and Comments:
1. Is an Intruder/User decision made at each LSTM time step (frame)? Please clarify.
2. I question the use of bi-directiona LSTMs for real-time tracking because they do not preserve causality. Please address this fundamental and critical issue.
Reviewer 2 Report
In this paper, an integral project is presented and verified by field test, which is a very interesting work. The presentation and layout of the paper are very good. I have only one concern. This project includes person identification, tracking, and classification tasks, it would be better to comparatively evaluate these three tasks with mainstream models.
Reviewer 3 Report
Readibility of the paper can be greatly improved.
There should be enough details that a reader wishing to repeat the experiments can do so, especially as regards the deep learning. I do not think that is the case here. It is a pity because results are most impressive.
The level of English is fine for the most part, but there are several places where the English must be significantly improved.
Reviewer 4 Report
1. It's better for the author to highlight the contribution of this paper. It seems the author use the matural method to solve this problem.
2. It's better for the author to add some algorithm description to better explain the algorithm.
3. In the section of system design, it's better for the author to add more details to explain the method more clearly.
4. Some of the reference is not in right format.
Round 2
Reviewer 1 Report
None
Reviewer 3 Report
I think the additions improve the quality of the paper. It can be published now.
Reviewer 4 Report
The authors has already answered the reviewer's comments.
It could be published in this version.